

# Assessing riverbank erosion in Bangladesh using time series of Sentinel-1 radar imagery in the Google Earth Engine

Jan Freihardt[1] & Othmar Frey[2,3]

[1]Centre for Comparative and International Studies (CIS), ETH Zurich, 8092 Zurich, Switzerland
[2]Institute of Environmental Engineering, ETH Zurich, 8093 Zurich, Switzerland
[3]Gamma Remote Sensing, 3073 Gümligen, Switzerland

*Correspondence to*: Jan Freihardt (jan.freihardt@ir.gess.ethz.ch)

**Abstract.** Riverbank erosion occurs along many of the Earth's river systems, affecting riverine populations by destroying agricultural land and housing. In this study, we detected past events of riverbank erosion along the Jamuna River in Bangladesh using time series of Sentinel-1 satellite radar imagery, ground range detected (GRD) data with a 12-day revisit cycle, available in the Google Earth Engine (GEE). Eroded land is detected by performing a land cover classification and by detecting land cover changes from vegetated areas before the monsoon to sand or water after the monsoon. Further, settlements are detected as persistent scatterers, and classified as eroded if they are located on eroded land. We found that with Sentinel-1 data, erosion locations can be determined already one month after the end of the monsoon, and hence potentially earlier than using optical satellite images which depend on cloud-free daylight conditions. Further, we developed an interactive GEE-based online tool allowing the user to explore where riverbank erosion has destroyed land and settlements along the Jamuna in five monsoon seasons (2015-2019). The source code of our implementation is publicly available, providing the opportunity to reproduce the results, to adapt the algorithm and to transfer our results to assess riverbank erosion in other geographical settings.

## 1 Introduction

In Bangladesh, located in one of the largest river deltas of the world, riverbank erosion is among the most drastic environmental processes in terms of yearly damage. Around 20 out of 64 districts in Bangladesh are prone to riverbank erosion, which consumes around 8.700 ha of land each year and thereby affects around 200.000 people by destroying their house and/or their agricultural land (Alam, 2017).

Large-scale erosion – whereby several hundred square meters of land can collapse into the river within short time – mainly happens during the rainy monsoon season typically from June to October. Such erosion events occur primarily in a limited number of hotspot areas along the three major streams of Bangladesh, Jamuna, Ganges and Meghna. Jamuna River is one of the largest braided river systems in the world, forming various channels at a total width of around 12 km. Since the 1970s, its bank line has shifted by around 20 km, continuously eroding the riverbank and creating new land (mainly in the form of islands).

Each year, the Bangladesh Water Development Board (BWDB) commissions an assessment of last year's erosion based on optical satellite imagery (e.g. CEGIS, 2018). This report is usually available only a few weeks before the beginning of the monsoon season, that is in May. This is due to the dependence of the analysis on cloud-free optical images, which are available in November/December for certain years, but only in January for other years. There is, thus, a need to establish an erosion assessment that is independent of cloud conditions and potentially available earlier, which would give communities along the rivers more time to prepare for the upcoming monsoon season.



In general, two distinct approaches exist to assess riverbank erosion quantitatively. First, the river system can be simulated using morphological numerical models. The capacities of such models increased significantly with the development of more powerful computers in the 2000s (Williams et al., 2016; Langendoen and Simon, 2008; Luppi et al., 2009). Computing power is necessary since fluvial systems are highly complex due to the large number of processes, scales and dimensions involved. Applying a numerical model to a river system as complicated as a braided river, however, would be extremely difficult, if not impossible.

The second approach to assess erosion at the large spatial scale of entire river systems is remote sensing, using either passive or active systems. Passive optical systems are widely used and serve a variety of purposes. One important application is the classification of land cover (Trianni et al., 2014; Du et al., 2016; Rishikeshan and Ramesh, 2018; Donovan et al., 2019; Immitzer et al., 2016). A second field of application is the monitoring of earth system processes, such as quantifying and mapping riverbank erosion and accretion along the Ganges (Hossain et al., 2013), the Yellow River (Chu et al., 2006), the Mekong (Kummu et al., 2008), and the Jamuna and Padma Rivers in Bangladesh (Islam, 2009). Lastly, they can also help to generate hazard and risk maps, for instance, for landslide hazard (Joyce et al., 2009) or flood risk (El-Behaedi and Ghoneim, 2018).

Passive optical systems rely on receiving reflected sunlight from the Earth's surface which leads to a significant drawback: They cannot image the Earth's surface at night or under cloudy conditions. While the former is problematic mainly for rapidly occurring events such as floods or storms, the latter can affect any application, especially in cloud-prone regions. For land cover classification or monitoring of slowly occurring phenomena such as glacier movement or land cover change, cloud coverage of individual images can usually be compensated by information from cloud-free images obtained at earlier or later times. Yet, this strategy does not work if cloud coverage is continuous for a prolonged period. This is the case in Bangladesh, where cloud coverage is high during the monsoon season lasting for months.

Active microwave sensors such as lidar and radar emit a signal themselves and measure the radiation that is reflected from the target. Today, the most important imaging radar technology used in remote sensing applications is Synthetic Aperture Radar (SAR) which provides high-resolution two-dimensional images independent from daylight, cloud coverage and weather conditions (Moreira et al., 2013).

Similar to optical systems, radar systems are employed in a wide range of applications. Examples include the extraction of shorelines (Al Fugura et al., 2011) and rivers (Sghaier et al., 2017), mapping of open water bodies (Santoro and Wegmuller, 2014), or land cover classification (Cable et al., 2014). On the topic of natural disasters, extensive research has investigated the use of radar for mapping the extent and depth of floods, for instance in the Amazon (Martinez and Le Toan, 2007), the USA (Townsend, 2001), Taiwan (Chung et al., 2015), as well as for monsoon flooding in Bangladesh (Imhoff et al., 1987). Further, several studies used SAR data for fully automated flood detection to provide near-real time disaster information (Martinis et al., 2009; Martinis et al., 2015; Twele et al., 2016). Thus, time series of spaceborne SAR images are potentially suitable to detect riverbank erosion and are, with Sentinel-1, available with short temporal sampling (6- or 12-days repeat-pass) on a continental to global scale.

In this paper, we present a feasibility study on riverbank erosion assessment based on time series of Sentinel-1 SAR imagery. Our study area is the Jamuna River in Bangladesh, for which a large-scale erosion assessment based on radar – and hence independent of cloud or weather conditions – has not been done yet. We employ a radar backscatter based detection of specific locations affected by riverbank erosion and we quantify their spatial extent for both eroded farmland and eroded settlements. The quality of the classification is evaluated with cloud-free Sentinel-2 optical data. We also



assess 1) the "time to detection" after the monsoon and 2) the spatial resolution of the erosion detection, both of which are crucial parameters for potential emergency response and damage assessments.

Riverbank erosion occurs not only along the Jamuna River and other rivers in Bangladesh but also along various major rivers worldwide (e.g. Mekong River, Yellow River, Mississippi River or Danube River). Given that the algorithm developed in this study is publicly available, it can potentially be transferred and adapted to other geographical settings at comparatively low effort and cost. This makes the findings of our study relevant beyond the specific case study of Bangladesh.

## 2 Methods and Data

### 2.1 The Google Earth Engine

A free and thus very attractive platform for analyzing remote sensing data is the Google Earth Engine (GEE). The GEE is a cloud-based platform providing access to a wide range of publicly available remote sensing data in connection with Google's massive cloud computing resources (Gorelick et al., 2017). The platform can be accessed free of charge by scientists, practitioners and other non-commercial users. Since its introduction in 2017, the GEE has been used for many remote sensing based (research) projects including applications close to the topical focus (e.g. mapping floods (Liu et al., 2018) or wetland dynamics (Muro et al., 2019)) or geographic focus of this work (e.g. monitoring rice growth in West Bengal (Mandal et al., 2018) or Bangladesh (Singha et al., 2019)).

Using the GEE is appealing since it gives simple access to a vast amount of remote sensing data, which do not have to be downloaded locally, but are processed in the cloud. Further, the GEE is relatively easy to use and does not require special software on the user side. Therefore, it can be applied in operational settings with limited resources, be it in terms of finances or trained personnel. GEE code can be shared conveniently via one link. The algorithm developed in this study can thus be easily accessed and adapted by research institutes or government authorities in Bangladesh. For all these advantages, this study used the GEE for all analyses.

Due to its computing architecture, the GEE can process only the amplitude, but not the phase information of radar images. The amplitude value corresponds to the reflectivity of an area, such that targets with high backscatter appear as bright spots in the radar image and flat smooth surfaces as dark (Moreira et al., 2013). As such, amplitude values can for instance be used to classify land cover. Due to this limitation of the GEE, the method presented subsequently works with backscatter coefficients only.

### 2.2 Data and pre-processing

This study used publicly available satellite imagery from the European Space Agency's (ESA) Sentinel mission (for more details see ESA, 2020b), launched in 2014, which collects C-band SAR images of the entire Earth's surface with a 6- to 12-day revisit cycle. Optical images were obtained from ESA's Sentinel-2 mission launched in 2015 with a 2- to 3-day revisit cycle at mid-latitudes (ESA, 2020c).

The Copernicus Sentinel-1 SAR data [2014-2021] used in this study was accessed through the GEE. The level-1 ground-range detected (GRD) scenes available in the GEE have already been pre-processed by the GEE following the steps from ESA's Sentinel-1 toolbox (Veci et al., 2014; Google Developers, 2020). Since Sentinel-1 collects SAR data at a variety of modes, polarizations and resolutions, the pre-processed images provided by the GEE were filtered before the analysis to create a homogenous subset of data:



- Acquisition mode: Interferometric Wide Swath (IW) mode was selected since it is the primary conflict-free mode
providing the 6- to 12-day revisit cycle over land (ESA, 2020a).
- Resolution: The IW images were filtered to keep only high-resolution images (pixel spacing of 10x10 m).
- Incidence angle: To reduce backscatter variation, only images with a look angle between 30° and 45° were kept.
- Look direction (ascending/descending): The influence of both look directions was tested for the detection of
settlements. For the land cover classification, the ascending orbit was chosen. The relative orbit number of all
ascending and descending images was 114 and 150, respectively, ensuring identical imaging geometry for all
images of a certain look direction. Per look direction, the revisit cycle was 12 days.
- Polarization: For the IW mode, VV and VH polarizations are available. Since VH is available only from 2017
on, all analyses were performed on VV images.

Figure 1 gives an overview of the steps taken to develop the erosion detection algorithm. Methodological details are
explained in the following sections. The full code used to develop the classifiers for land cover and settlements can be
found here: https://code.earthengine.google.com/eb09ee5f4635c72daca0a93d085cdbbe

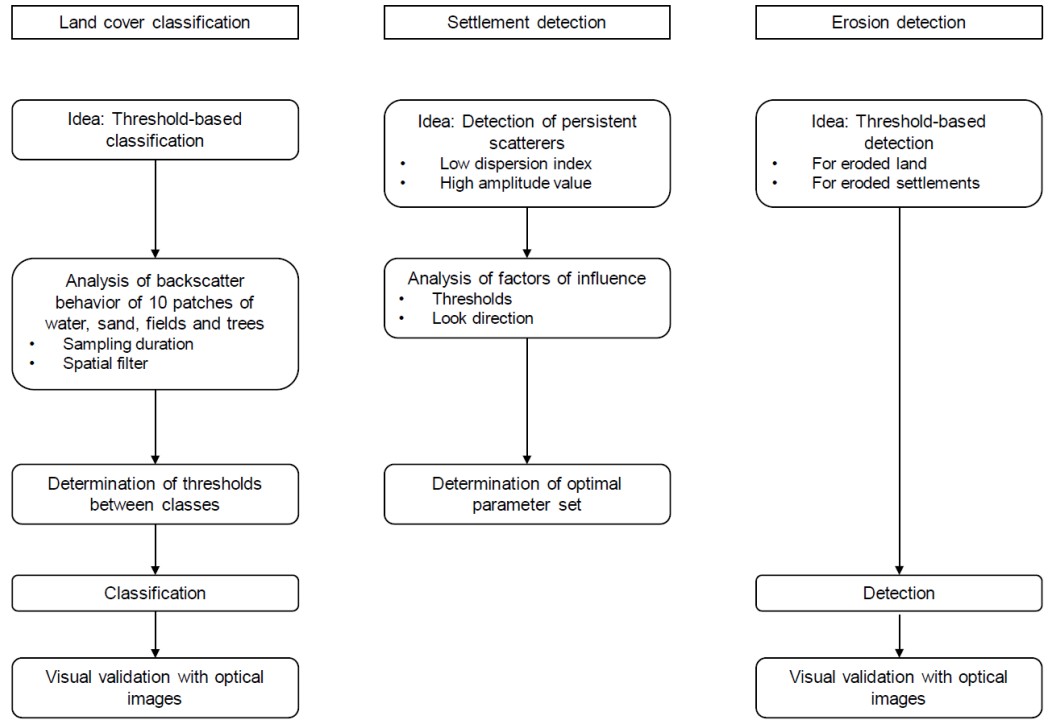

**Figure 1: Overview of the analytical strategy to develop an algorithm detecting eroded land and settlement.**

**2.3 Land cover classification**

To get a visual impression of the backscattering characteristics of different land cover types, the average backscatter
coefficient of five classes (settlement, trees, fields, sand and water) was plotted for the period from January 2018 to
February 2020. For each of the four land cover classes water, sand, trees and agricultural fields, ten patches of size
100x100 m were chosen based on visual inspection of the optical satellite images provided by the GEE. For each class,
the ten patches were distributed along the length of the Jamuna River. The locations of the patches are shown in Fig. A1.

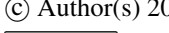



The speckle inherent in SAR images can be reduced by temporal averaging (maintaining spatial resolution, but requiring several images) or by spatial filtering (requiring only one image, but reducing spatial resolution). There exists, thus, a tradeoff between keeping full spatial resolution and using only a few images. To assess riverbank erosion in Bangladesh, it would be ideal to use only a few images (to obtain the assessment as early as possible after the end of the monsoon season) while maintaining spatial resolution (to have a precise estimate of the erosion extent). Therefore, a compromise
has to be found between sampling duration and spatial resolution. We tested the influence of these two parameters for a range of configurations:

- Eight sampling durations: two weeks; 1 month; 2, 3, 4, 5, 6, 7 months. Each of these eight periods started on 01 November 2018. All images within the respective period were averaged temporally before the subsequent analysis.

- Seven spatial filters: no filter; 3x3 refined Lee filter; 3x3, 5x5, 7x7, 25x25 and 50x50 boxcar filter (Lee, 1981; Lee et al., 2009). The filters were applied to the absolute backscatter values.

For a certain imaging configuration (sampling duration and filter type), the mean backscatter as well as the standard deviation of the pixels within each patch were calculated. Subsequently, these ten patch-specific mean and standard deviation values were averaged to yield one mean backscatter and one standard deviation value per land cover class and
imaging configuration.

To classify pixels into one of the four classes, thresholds were defined between water/sand, sand/fields and fields/trees. The thresholds were calculated as $0.5 * \left[ \left( mean_i + n * \sigma_i \right) + \left( mean_j - n * \sigma_j \right) \right]$ where $i$ and $j$ indicate the class with the lower and higher mean backscatter, respectively. $n$ was chosen as the largest natural number such that $\left( mean_i + n * \sigma_i \right)$ and $\left( mean_j - n * \sigma_j \right)$ would not overlap. $n$ could thus be different for each pair of classes. For trees, an additional upper
threshold was set at -2 dB to distinguish them from settlements. Pixels were classified according to their backscatter value with respect to these thresholds. For instance, a pixel with a backscatter value larger than the threshold water/sand, but smaller than the threshold sand/fields was classified as "sand". The quality of the classification was assessed visually using optical Sentinel-2 images.

**2.4 Settlement detection**

Since houses in rural Bangladesh are typically surrounded by trees, they are not fully visible on satellite images. Moreover, they cover only small areas compared to water, sand or farmland. Therefore, they cannot be well detected with the classification approach presented in Sect. 2.3, which involves spatial averaging.

To detect houses, we exploit the fact that unlike vegetation, houses do not move or change substantially over time. Due to this low temporal decorrelation, houses are treated as persistent scatterers (PS) (Ferretti et al., 1999). Detecting PS
candidates in radar images usually implies analyzing phase coherence, which cannot be done in GEE where only amplitude information is available. However, Ferretti et al. (2001) show that phase dispersion can be estimated from the amplitude dispersion index $\sigma_A/m_A$ where $m_A$ and $\sigma_A$ are the mean and the standard deviation of the amplitude values, respectively. PS can then be selected by computing the dispersion index of each pixel from a stack of several SAR images of the same scene and keeping only those pixels exhibiting a low dispersion index. The typical range of threshold values
for the dispersion index goes from 0.25 (Ferretti et al., 2001) to 0.4 (van Leijen, 2014).

Houses are not the only structures than can have a low dispersion index. Bare surfaces, for instance, might also be relatively stable over time. Therefore, we combine the dispersion criterion with an amplitude threshold: Pixels are selected as PS candidates and hence houses if they show a low dispersion index and a high absolute backscatter over a series of





radar acquisitions. Two implementations of the amplitude threshold were compared: First, following Kampes and Adam

(2004), a pixel is selected as PS candidate if its normalized cross section $\sigma_0$ is above a threshold $N_2$ in at least $N_1$ images. These authors propose thresholds of -2 dB for $N_2$ and 0.65$K$ for $N_1$, where $K$ is the number of radar acquisitions. Second, the amplitude threshold was applied to the mean of all SAR images in the stack, instead of the individual images.

The sensitivity of the settlement detection was tested for the following parameters:

- Thresholds: For the dispersion index, threshold values of 0.25 and 0.4 were tested. For the amplitude threshold,
180        -4 dB, -2 dB and 0 dB were analyzed. These analyses were done for a sampling duration of six months starting 01 November 2019.
- Look direction: Since the roofs of buildings typically have a specific orientation, they are likely to have a stronger backscatter for one of the two look directions "ascending" or "descending". Therefore, these two types were compared.

**2.5 Erosion detection**

In examining the impact of riverbank erosion on human livelihoods along the Jamuna River, we are interested in two effects, which are treated separately: erosion of land (farmland or trees) and erosion of houses. Land was identified as eroded in one specific monsoon season if it was classified as "field" or "trees" before the monsoon season and as "sand" or "water" afterwards. Erosion to sand and erosion to water were not differentiated further since in both cases, the land

cannot be used for agriculture anymore, which is the main effect we are interested in in this application.

For classifying the land, a sampling duration of six months (November to April) was used for all years from 2014/15 to 2018/19. For 2019/20, only the images from November 2019 were used to simulate the case that the erosion detection has to be performed already in December after the end of the monsoon. A 7x7 boxcar filter was applied to create smooth and continuous erosion bands. The threshold discriminating sand/water from fields/trees was -13.2 dB and -12.7 dB for

the case where six months and one month of data were used, respectively (Table B1 and Table B2). To detect eroded settlements, a similar strategy was followed: A pixel was selected as settlement eroded during a specific monsoon season if it was classified as "settlement" before the monsoon and as "sand" or "water" afterwards. The final algorithm to detect eroded land and settlement is schematized in Fig. 2.



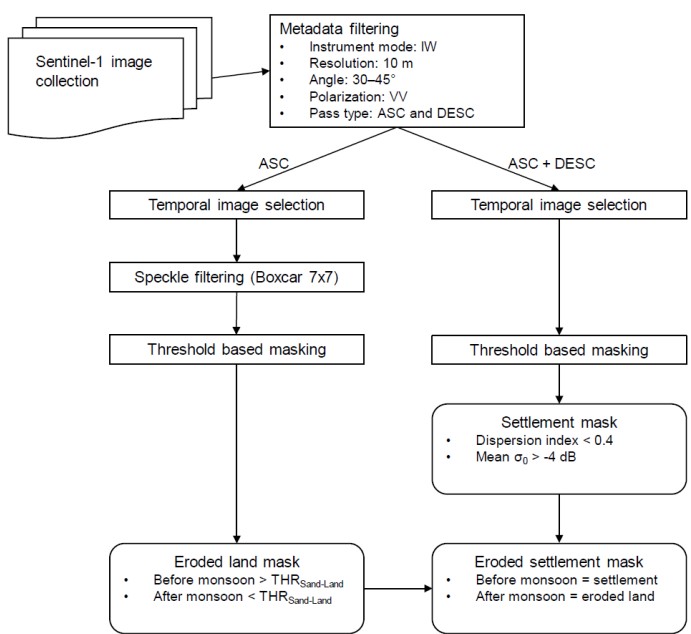

**Figure 2: Flow chart of the final implementation to detect eroded land and settlement. THR = threshold.**

### 2.6 Accuracy assessment

Since our study area is large (spanning more than 200 km from north to south), it was not feasible to collect sufficient ground-truth data to validate our land cover classification algorithm. To still be able to judge its accuracy, we compared the SAR-based classification to an independently conducted classification based on optical Sentinel-2 imagery, performed in the GEE.

Classification of Sentinel-2 images was based on the Normalized Difference Vegetation Index (NDVI). The NDVI takes on values between -1 and 1. In terms of the land cover classes relevant for our study, water bodies typically exhibit NDVI values below 0, bare ground between 0 and 0.1, and cultivated land above 0.1 (DeFries and Townshend, 1994; Huang et al., 2020).

Since the detection of eroded land relies only on the threshold between sand and vegetation (cf. Sect. 2.5), we differentiated only two landcover classes in the Sentinel-2 classification: sand/water (corresponding to all pixels exhibiting an NDVI value < 0.1) and vegetation/trees (corresponding to all pixels exhibiting an NDVI value > 0.1). While this is a large simplification, it serves the purposes of this study where we try to detect land that changes from vegetated before the monsoon to sand or water after the monsoon.

The pixel-level accuracy of the SAR-based classification was assessed for one site by counting all pixels which were a) identically classified as vegetation by both methods, b) "false positives" (classified as vegetation by the SAR-based method, and as sand/water by the Sentinel-2 based method), and c) "false negatives" (classified as sand/water by the SAR-based method, and as vegetation by the Sentinel-2 based method), respectively.

The quality of Sentinel-2 images depends on cloud cover. In Bangladesh, cloud cover varies seasonally, with highest values occurring during the monsoon (June to September) and lowest values in the dry season (November to March) (Fig. A2). Therefore, the accuracy assessment was performed for the two months with the lowest cloud cover (November and



March), using the one-month median NDVI value, respectively. For both November and March, the accuracy assessment was repeated for three consecutive years (2018/19 to 2020/21). The code for the Sentinel-2 based assessment is contained in the GEE code referenced in Sect. 2.2.

**3 Results**

**3.1 Land cover classification**

The average monthly backscatter of seven patches is shown as a time series in Fig. 3. The settlement and tree patch are the most stable since they are neither affected by (rapid) vegetation growth nor by monsoon flooding. The river patch has mostly the lowest coefficient, which increases during the monsoon, potentially due to wind and rain disturbing the flat

water surface. While fields generally have a backscatter coefficient close to that of trees, they can be seasonally flooded during the monsoon (field 2) or completely eroded (field 3). From the behavior of field 2, the dry season can be defined as the period between November and June of the following year (indicated by the vertical lines). The sand patch lies in between water and fields.

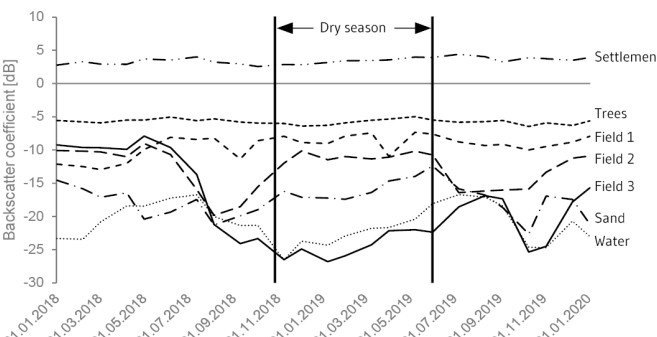

**Figure 3: Mean monthly backscatter of different land cover types (one patch per type; the location of the patches is shown in Fig. A1) between January 2018 and February 2020, obtained from C-band of Sentinel-1. Per month, between one and six images are averaged.**

The average backscatter of the ten sand patches is shown in Fig. 4a as a function of different sampling durations and filter configurations. For a given filter, there is no significant variation of the backscatter value with increasing sampling

duration. For a given sampling duration, the average backscatter increases slightly with increasing filter size. However, this increase becomes statistically significant at the 95 % level only for the largest filters (25x25 and 50x50 pixels). For such large filters (50x50 pixels corresponds to 500x500 meters), this is probably caused by other land cover classes with a higher backscatter value (e.g. fields) being included into the filter window.



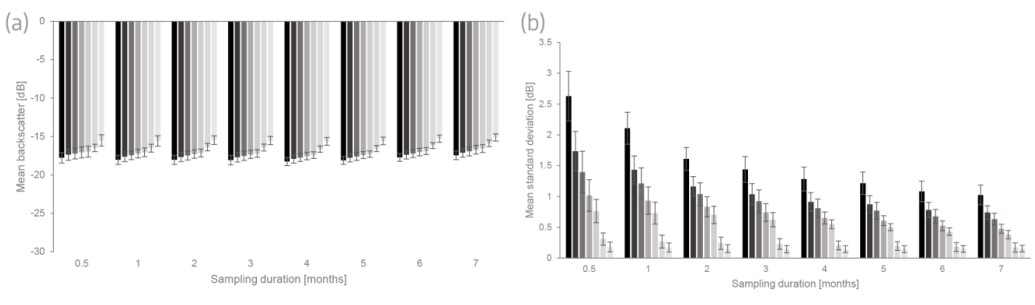


**Figure 4: Average backscatter (panel (a)) and average standard deviation (panel (b)) of the pixels within ten patches of sand for different sampling durations and filter sizes. Bars indicate the 95 % confidence interval. Lee – Lee filter. Box – boxcar filter.**

Figure 4b presents the standard deviation of all pixels within one patch, averaged over the ten sand patches. For a given

filter, the standard deviation decreases with increasing sampling duration. For a given sampling duration, it decreases with increasing filter size. These observations correspond to the two mechanisms for speckle reduction outlined in Sect. 2.3, namely temporal averaging and spatial filtering. The other three land cover classes "water", "fields" and "trees" show similar tendencies for filter size and sampling duration, both for average backscatter values and standard deviations (Fig. A3).

These findings allow defining thresholds to separate the four classes in the land cover classification. As discussed in Sect. 2.3, each combination of sampling duration and filter size has a certain advantage and a certain disadvantage. To illustrate this tradeoff, two extreme combinations are compared in Fig. A4. In practice, a compromise between these two extremes seems more likely, meaning that some spatial resolution has to be given up when a slightly longer sampling duration is used. One example for such a compromise is presented in Fig. 5, for which images from one month have been filtered

with a 3x3 boxcar filter.

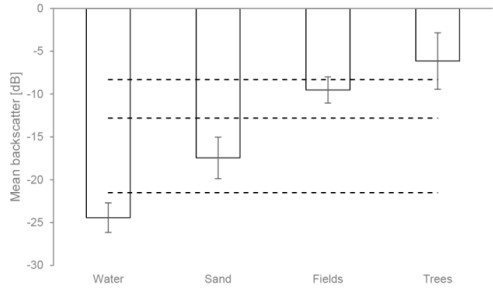

**Figure 5: Average backscatter for four land cover classes for a sampling duration of one month and a 3x3 boxcar filter. Bars indicate the mean ± 2 standard deviations. Horizontal lines indicate the thresholds between the respective classes.**

The determination of the thresholds is illustrated in Table 1 for the case of one month sampling duration and a 3x3 boxcar

filter. As can be seen in Fig. 5, the bars of fields and trees overlap if two standard deviations are used. However, they do not overlap if only one standard deviation is used. Hence, intervals with one standard deviation are used to determine the threshold between fields and trees. For water/sand and sand/fields, the intervals do not overlap even if three standard deviations are considered. Therefore, three standard deviations are used to calculate the respective thresholds. The thresholds for the two configurations from Fig. A4 are contained in Table B3 and Table B4. The average backscatter




values shown in Table 1, Table B3 and Table B4 compare reasonably well to reference values from the literature (Table B5).

**Table 1: Determination of thresholds for a sampling duration of one month and a 3x3 boxcar filter. Values in bold are those that have been used to calculate the threshold indicated in the last column. All values are in dB. μ – mean. σ – standard deviation.**

|        | μ     | σ   | μ+σ   | μ-σ   | μ+2*σ | μ-2*σ | μ+3*σ | μ-3*σ | Threshold |
|--------|-------|-----|-------|-------|-------|-------|-------|-------|-----------|
| Water  | -24.4 | 0.9 | -23.6 |       | -22.7 |       | **-21.8** |       |       |
| Sand   | -17.4 | 1.2 | -16.2 | -18.6 | -15.0 | -19.9 | **-13.8** | **-21.1** | -21.5 |
| Fields | -9.5  | 0.8 | **-8.8** | -10.3 | -8.0  | -11.0 | -7.2  | **-11.8** | -12.8 |
| Trees  | -6.1  | 1.7 |       | **-7.8** |     | -9.4  |       | -11.1 | -8.3  |


Assuming the backscatter values in each class to be distributed normally around the mean, this approach allows an estimation of the accuracy of the resulting classification. In a normal distribution, 68 %, 95 % and 99.7 % of all values lie within "mean ± one, two and three standard deviations", respectively. As the thresholds "water-sand" and "sand-fields" are based on the interval with three standard deviations, we thus expect less than 0.15 % of all water pixels to be incorrectly

classified as sand pixels. The same percentage applies for sand pixels being incorrectly classified as water/field pixels and for field pixels being misclassified as sand pixels. For "fields-trees", only one standard deviation has been used, and hence 16 % of all field/tree pixels are expected to be falsely classified as tree/field pixels, respectively.

Trees and fields can thus not be well distinguished in this setup, This shortcoming is, however, negligible in the context of studying riverbank erosion. Here, the focus is on land covered by fields or trees being eroded and appearing as sand or

water afterwards. Therefore, the most important threshold is the one between sand and fields, which yields higher accuracy.

The classification resulting from these three imaging configurations is depicted in Fig. 6 with an optical Sentinel-2 image as the baseline. If only two weeks are sampled with a 25x25 boxcar filter, the spatial resolution is largely lost (top right). If, by contrast, no filter is applied and six months are sampled, the classification remains very fine-grained (bottom left).

However, the distinction between sand and water is not very accurate. The compromise – one month sampling duration and a 3x3 boxcar filter (bottom right) – manages to preserve a large degree of spatial resolution while distinguishing well between the four classes. It thus seems the most appropriate of these three imaging configurations.





**Figure 6: (a) Sentinel-2 image of a stretch of the eastern riverbank of Jamuna River, taken in November 2019. Image dimensions: ca. 4.5x6 km. (b), (c) and (d) Classification result for a sampling duration of two weeks/25x25 boxcar filter, six months/no filter and one month/3x3 boxcar filter, respectively. Blue – water, sand – sand, light green – fields, dark green – trees. Source of optical background image: Sentinel-2. The location of the patch is shown in Fig. A1 (patch 1). Coordinates in this and all other maps are in "Gulshan 303 Bangladesh TM" (EPSG 3106).**






### 3.2 Settlement detection

Figure A5a illustrates the need to apply an amplitude threshold in addition to the dispersion index. If only the dispersion index is used to classify settlements, many vegetation pixels that happen to be stable over the sampled time window are misclassified as settlements. The influence of the dispersion index threshold and the amplitude threshold is shown in Fig. 7. For the dispersion index, a threshold of 0.25 (panel (b)) selects only very stable pixels as PS candidates. Accordingly, less pixels are selected than for a threshold of 0.4 (panel (a)). For the amplitude threshold, the effect is the

opposite: Applying a threshold of -4 dB (red) selects more pixels as PS candidates than for -2 dB (blue) or 0 dB (orange). While the threshold of -4 dB thus seems to select more settlement pixels (e.g. upper left corner of panel (a)), also the risk of misclassifying tree pixels as settlements rises.

For our application, however, we are rather interested in detecting the rough location of settlements than in precisely distinguishing settlement and tree pixels. In fact, trees are often planted around houses, making them good indicators of

settlements. As too few settlement pixels are detected in panel (b), we suggest a threshold combination of 0.4 and -4 dB for the dispersion index and the amplitude, respectively. Still, it is important to note that even with this combination (red pixels in panel (a)), several pixels that appear as houses in the optical image are not detected. We should thus keep in mind that what the algorithm classifies as settlement is most likely indeed a settlement, but that it cannot detect all settlements, especially if they are covered by trees.

The influence of the look direction is illustrated in Fig. A5b. The overlap between the ascending and descending orbit is small. This corresponds to the fact that each building has a specific orientation of its roof. Therefore, some roofs have a stronger backscatter in the descending orbit, while others reflect more in the ascending orbit. Similar effects can be expected if persistent scatterers are located on walls or in corners of buildings. For maximum settlement detection, it is thus recommended to use SAR images of both look directions. To conclude, the recommended set of parameters to detect

settlements is to use an amplitude and dispersion index threshold of -4 dB and 0.4, respectively, using images of both ascending and descending orbit.

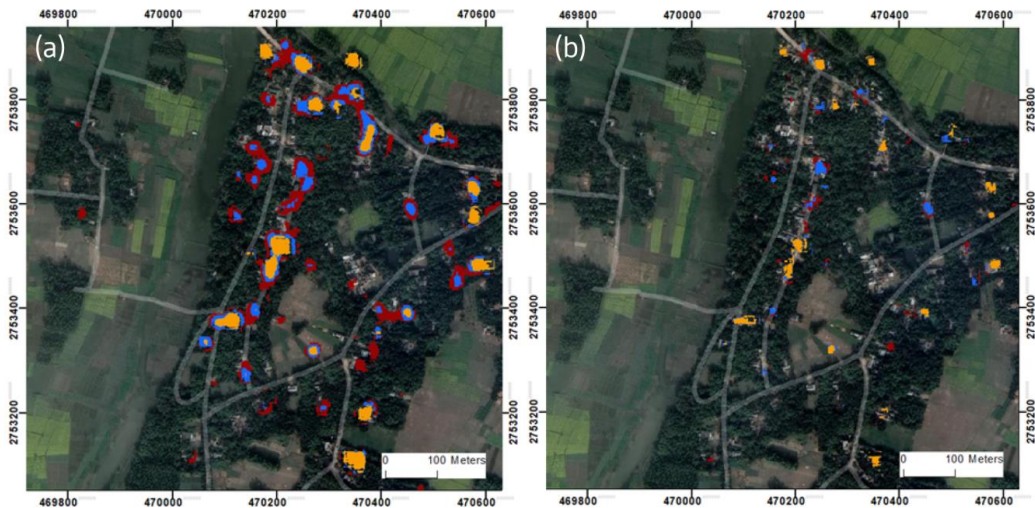

**Figure 7: Influence of classification thresholds on settlement detection. Shown are dispersion index thresholds of 0.4 (panel (a))
and 0.25 (panel (b)) – the lower the threshold, the more stable a pixel has to be for it to be classified as a PS candidate. Colors**
**correspond to different values of the amplitude threshold: -4 dB (red), -2 dB (blue), 0 dB (orange) – the lower the threshold,**



**the higher the chance of classifying tree pixels as settlement pixels. Source of optical background image: Google, ©2020 Maxar Technologies, CNES/Airbus.**

### 3.3 Erosion detection

Figure 9 illustrates the result of the erosion detection (both land and settlements) for one specific site for the monsoon seasons 2018 and 2019. To evaluate the quality of the erosion detection, the detected erosion patches are mapped on optical images from before and after the monsoon. The results of the overall validation of the land cover classification are contained in Sect. 3.4.

The focus of the project is on erosion occurring on the outer riverbanks of the Jamuna. Therefore, erosion happening on the sandbanks and islands in the river is omitted in the following discussion, which focuses exclusively on the long strip of eroded land on the outer riverbank. For both years, all that has been detected as eroded land has entirely been land before the monsoon (left column) and completely water after the monsoon (right column). For these examples, there is thus no type I error, i.e. classifying land as eroded when it is not.

There is, however, a type II error, i.e. eroded land that is not classified as such. This error tends to be small and thus negligible for the overall purpose of detecting sites where erosion occurred to a significant extent. Lastly, the algorithm can distinguish well between eroded land and eroded sand, as can be seen in the lower left corner of the 2019 image before the monsoon. Regarding the patches detected as eroded settlements (bright red), by far not all of the eroded settlement is detected. Again, this type II error is negligible given the purpose of detecting those sites that have seen erosion of settlement in general. For this, it is not necessary to detect every single house that has been eroded.

The erosion detection works for the monsoon seasons from 2015 to 2019, since Sentinel-1 images are only available from October 2014 onwards. Figure 9a shows the sequential nature of erosion, which does not occur at random locations, but typically in sites which have already experienced erosion during the previous monsoon season(s). We can also see the highly dynamic nature of land accretion and erosion. For instance, an island had formed at the place where land had been before the 2015 monsoon. Part of this island has been eroded again in the 2019 monsoon season (orange patch overlaying the dark brown 2015 erosion band). Further, settlements have been eroded in all five monsoon seasons (blue dots). Figure 9b shows where land was eroded in the 2019 monsoon along a larger section of the Jamuna River. Erosion occurred within, but to a large extent also outside of the hotspot areas predicted by CEGIS (2018).





**Figure 8: Validation of erosion detection. Shown are eroded land (orange) and eroded settlements (red) for 2018 ((a) and (b)) and 2019 ((c) and (d)). Baseline: optical Sentinel-2 images from before ((a) and (c)) and after the monsoon ((b) and (d)). The location of the patch is shown in Fig. A1 (patch 3).**

355



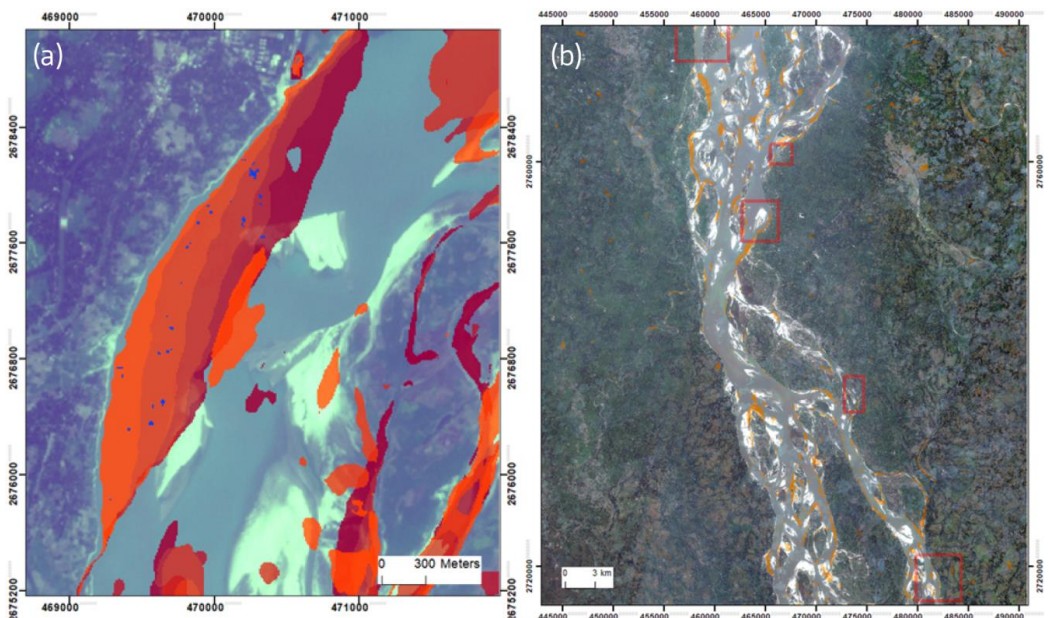

**Figure 9: (a) Detected erosion for the monsoon seasons 2015 (dark red) to 2019 (light red). Blue: eroded settlements. (b)**
**Locations of land eroded during the 2019 monsoon (orange). The red rectangles are the locations for which CEGIS (2018)**
**predicted severe erosion. Source of optical background images: Sentinel-2.**

### 3.4 Accuracy assessment

A confusion matrix of vegetation versus sand/water was calculated for six different months (Table B6). The accuracy
metrics were averaged over three consecutive years for both November and March (Table 2). The classification showed
a satisfactory accuracy over 87 %. The observed errors might be introduced by the cloud mask which is applied to the
Sentinel-2 images and potentially leads to misclassified pixels.

**Table 2: Accuracy assessment of the SAR-based classification for two months. Indicated are the accuracy values averaged for**
**November/March over three consecutive years, respectively.**

| Month | Class | User's accuracy [%] | Producer's accuracy [%] | Overall accuracy [%] |
|-------|-------|---------------------|-------------------------|----------------------|
| November | Sand/water | 94.7 | 83.1 | 87.5 |
| | Vegetation | 80.1 | 93.8 | |
| March | Sand/water | 87.5 | 93.6 | 90.6 |
| | Vegetation | 93.7 | 88.1 | |

As mentioned in Sect. 2.6, a validation based on ground-truth data was not possible due to the large size of our study area.
Readers and users are, however, invited to access the source code in the GEE and compare the SAR-based land cover
classification to optical imagery for specific sites of interest.

### 3.5 Final implementation

Finally, the learnings from Sect. 3.1 to 3.3 were implemented in a GEE-based analysis tool that allows the user to explore
where erosion of land and settlement has occurred during the five monsoon seasons from 2015 to 2019. The tool contains
the following information:





- Five layers for land eroded in the five monsoon seasons, 2015 to 2019
- Five layers for settlements eroded in the five monsoon seasons, 2015 to 2019
- One layer for the settlement detected in the beginning of 2020
- Three optical images from January 2018, 2019 and 2020 as a visual baseline
- The 14 "erosion hotspots" identified by CEGIS in their 2019 erosion prediction (cf. Sect. 1)

The GEE-based tool to assess riverbank erosion using Sentinel-1 data can be accessed here:

https://code.earthengine.google.com/3ea8f1fd5d771accc621550d744a914e?hideCode=true

To introduce users who are unfamiliar with the GEE into the application of this tool, we have recorded a short tutorial:
https://youtu.be/_b9AAPDw7Wk

## 4 Discussion

Different limitations might affect the results of this study. First, as outlined in Sect. 2.1, the GEE contains only the amplitude, but not the phase values of radar images. The phase value contains information on the distance between the sensor and the ground, accurate to a small fraction of the radar wavelength. One powerful technique employing the phase
value is SAR interferometry which compares for one scene the phase of two or more radar images acquired from different positions or at different times (Moreira et al., 2013). Accessing the phase information could thus open up alternative strategies to detect eroded land, for instance from phase decorrelation. SAR interferometry, however, requires special software, which might not be available in resource-constrained settings. The GEE, by contrast, is easy and free to use, making the developed algorithm accessible to authorities and researchers in Bangladesh.

Second, our study used only radar data. Combining optical and SAR data generally yields an improved performance compared to using any of the two alone. Examples using both data types include land cover classification (Carrasco et al., 2019; Miettinen et al., 2019; Poortinga et al., 2019; Zhang et al., 2018), change detection (Canty and Nielsen, 2017; Celik, 2018; Shimizu et al., 2019) and the derivation of river discharge for the Upper Brahmaputra River (Huang et al., 2018). The GEE facilitates the combination of optical and SAR data. Such a combination would thus be another strategy
to further improve the results of this study.

Third, we have developed the algorithm to detect riverbank erosion for one specific case study. As it is usually the case for case study research, it is not evident how well our findings can be transferred to other contexts beyond Bangladesh. Given, however, that the basic mechanism of riverbank erosion (vegetated/settlement land turning into sand/water) is identical irrespective of where erosion occurs, we believe that it is possible to apply our erosion detection approach to
other contexts at relatively low effort (e.g. by adapting classification thresholds to local vegetation/soil types). We invite interested readers – both from research and from practice – to access our algorithm and to apply it to other geographical settings.

## 5 Conclusions

We have implemented and applied a GEE-based method to quantitively assess riverbank erosion along the Jamuna River
in Bangladesh based on Sentinel-1 GRD intensity data. Timely detection of riverbank erosion is an essential element of disaster risk management, yet especially challenging in resource-limited settings.

We investigated whether the locations of past erosion events can be extracted from Sentinel-1 SAR imagery. We developed an algorithm to classify land cover, identify settlements and detect eroded land and settlements along the Jamuna River. The SAR-based classification approach can provide information on where land and settlements have been



eroded during the last monsoon already one month after the end of the monsoon season, and hence potentially earlier or at least more reliably than using optical satellite images, which depend on cloud-free conditions. This erosion detection can be achieved at sufficiently high spatial resolution. We could thus demonstrate the suitability of radar imagery to assess past erosion events.

The analysis was performed using the GEE which gives access to Google's cloud computing infrastructure as well as to
massive amounts of satellite imagery, including time series of Sentinel-1 GRD backscatter data and Sentinel-2 optical data on a global scale. A limitation of using the GEE is that it contains only the amplitude-, and not the phase-values of the radar images. Landcover-change classification approaches using interferometric coherence can thus not be implemented in the GEE. However, the GEE facilitates sharing and re-using algorithms, making the results of this study accessible and useable for government agencies or NGOs in Bangladesh. To share our results, we developed an interactive
online tool allowing the user to explore where land and settlement have eroded along the Jamuna River in the monsoon seasons 2015 to 2019. This online tool as well as the underlying source code can be accessed and adapted free of charge, making it an attractive tool to use in resource-constrained settings.

Spatio-temporally consistent sequences of progressive riverbank erosion give valuable insights on where erosion will likely occur in the following monsoon season. Such information can be used to alert potentially affected residents
accordingly. As such, the code and tool developed in this study might be of interest to both policymakers and practitioners working in the fields of disaster risk management and communication. Since riverbank erosion is a phenomenon occurring along many of the world's major rivers, the relevance of our tool extends beyond the specific case study of Bangladesh. Likewise, it might be applicable to coastal erosion – another environmental hazard that is bound to increase in the age of climate change.




**Appendix A: Additional figures**



**Figure A1: (a) Locations of the patches shown in Fig. 3 (symbols are larger than the patches). (b) Locations of the patches analyzed for the development of the land cover classification (symbols are larger than the patches). (c) Locations of the patches used to validate the land cover classification (patch 1), the settlement detection (patch 2) and the erosion detection (patch 3). Exact coordinates for all patches are contained in the GEE source code. Source of optical background image: Sentinel-2.**



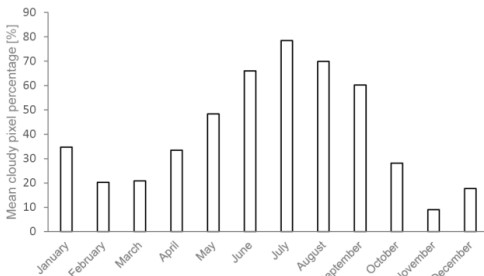

**Figure A2: Mean cloudy pixel percentage of all Sentinel-2 images taken over the assessment site during the respective month.**
**For each month, five consecutive years were analysed. The plotted values represent the average of these five years.**

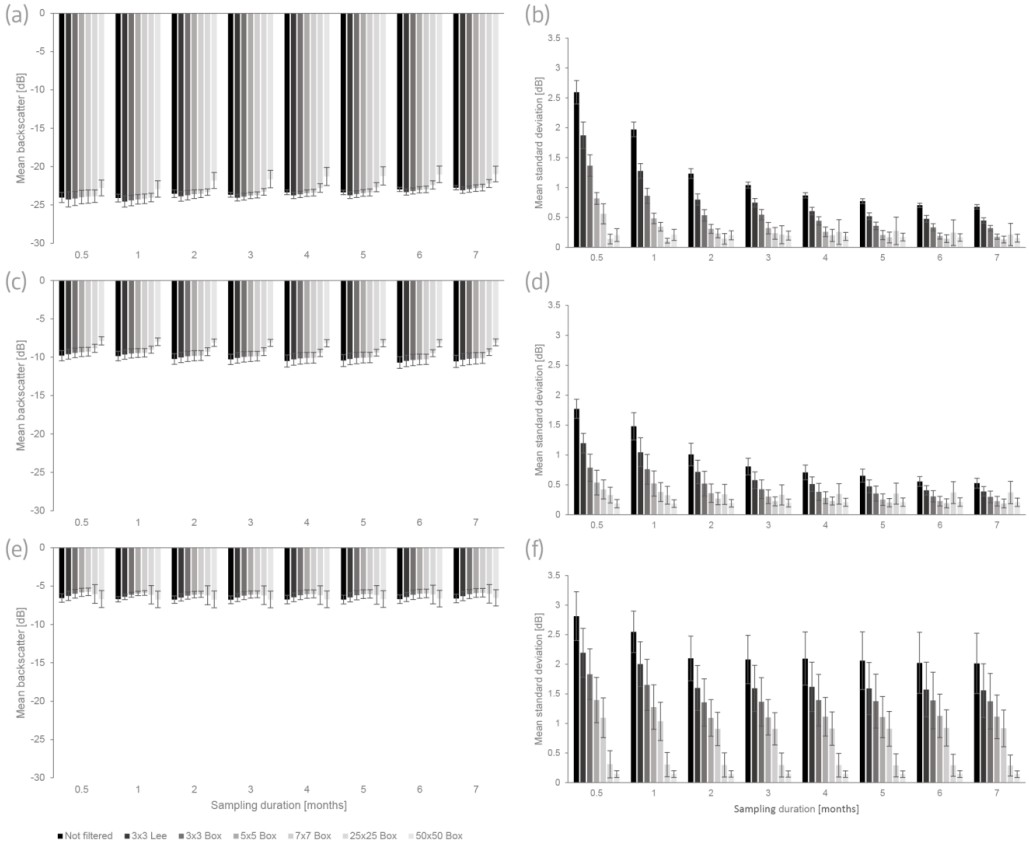

**Figure A3: Average backscatter (panels (a), (c) and (e)) and average standard deviation (panels (b), (d) and (f)) of the pixels within ten patches of water (panels (a) and (b)), fields (panels (c) and (d)) and trees (panels (e) and (f)) for different sampling durations and filter sizes. Bars indicate the 95 % confidence interval. Lee – Lee filter. Box – boxcar filter.**



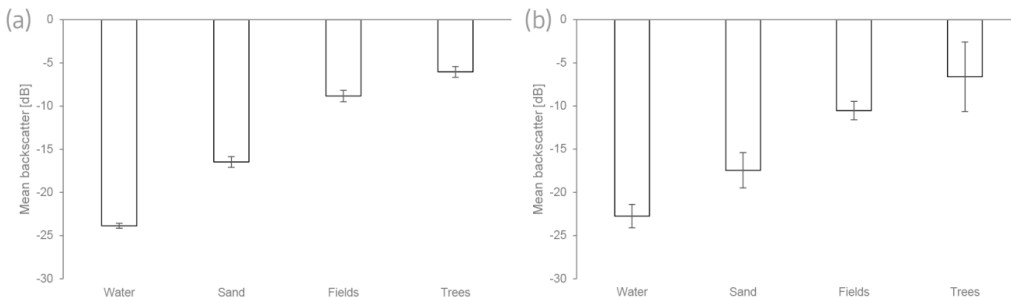


**Figure A4: Average backscatter for four land cover classes (a) for a sampling duration of 0.5 months and a 25x25 boxcar filter and (b) for a sampling duration of 7 months and no filter. Bars indicate the mean ± 2 standard deviations. If images are available only from two weeks, strong spatial filtering (25x25 pixels) reduces the standard deviation enough to separate all four classes even at the level of two standard deviations around the mean (panel (a)). If, by contrast, images are available from seven**
**months, water, sand and fields can be separated even if no spatial filter is applied (panel (b)). In this setting, fields and trees can be distinguished only at the level of one standard deviation around the mean (not shown in the graph).**

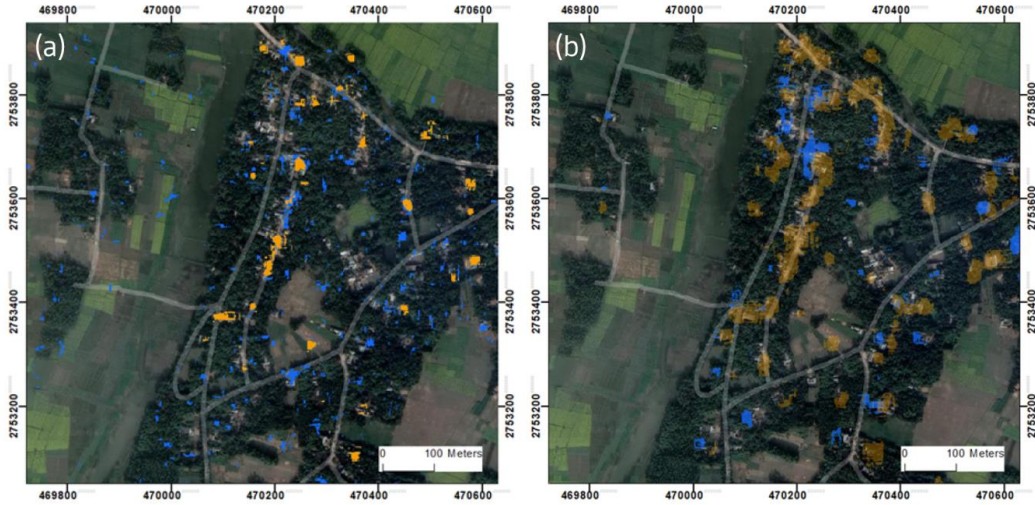

**Figure A5: (a) Classification of settlements using the dispersion index only (blue) or the combination of dispersion index and amplitude threshold (orange). Thresholds: 0.25 (dispersion index), -2 dB (amplitude). (b) Settlement detection for ascending**
**(orange) and descending (blue) orbit. Thresholds: 0.4 (dispersion index), -4 dB (amplitude). Sampling duration for both panels: six months. The location of the patch is shown in Fig. A1 (patch 2). Source of optical background image: Google, ©2020 Maxar Technologies, CNES/Airbus.**



**Appendix B: Additional tables**

**Table B1: Determination of thresholds for a sampling duration of six months and a 7x7 boxcar filter. Values in bold are those that have been used to calculate the threshold indicated in the last column. All values are in dB. μ – mean. σ – standard deviation.**

|        | μ     | σ   | μ+σ    | μ-σ    | μ+2*σ  | μ-2*σ  | μ+3*σ  | μ-3*σ  | Threshold |
|--------|-------|-----|--------|--------|--------|--------|--------|--------|-----------|
| Water  | -23.0 | 0.1 | -22.8  |        | -22.7  |        | **-22.5** |        |           |
| Sand   | -16.9 | 0.4 | -16.5  | -17.3  | -16.0  | -17.7  | **-15.6** | **-18.2** | -20.4  |
| Fields | -10.3 | 0.2 | -10.1  | -10.5  | -9.9   | -10.6  | **-9.7** | **-10.8** | -13.2  |
| Trees  | -6.0  | 0.9 |        | -6.9   |        | -7.8   |        | **-8.7** | -9.2    |

**Table B2: Determination of thresholds for a sampling duration of one month and a 7x7 boxcar filter. Values in bold are those that have been used to calculate the threshold indicated in the last column. All values are in dB. μ – mean. σ – standard deviation.**

|        | μ     | σ   | μ+σ    | μ-σ    | μ+2*σ  | μ-2*σ  | μ+3*σ  | μ-3*σ  | Threshold |
|--------|-------|-----|--------|--------|--------|--------|--------|--------|-----------|
| Water  | -24.2 | 0.3 | -23.9  |        | -23.5  |        | **-23.2** |        |           |
| Sand   | -17.1 | 0.7 | -16.3  | -17.8  | -15.6  | -18.5  | **-14.9** | **-19.2** | -21.2  |
| Fields | -9.4  | 0.4 | -9.0   | -9.8   | **-8.7** | -10.2 | -8.3   | **-10.6** | -12.7  |
| Trees  | -5.9  | 1.0 |        | -6.9   |        | **-8.0** |        | -9.0   | -8.3    |

**Table B3: Determination of thresholds for a sampling duration of two weeks and a 25x25 boxcar filter. Values in bold are those that have been used to calculate the threshold indicated in the last column. All values are in dB. μ – mean. σ – standard deviation.**

|        | μ     | σ   | μ+σ    | μ-σ    | μ+2*σ  | μ-2*σ  | μ+3*σ  | μ-3*σ  | Threshold |
|--------|-------|-----|--------|--------|--------|--------|--------|--------|-----------|
| Water  | -23.9 | 0.1 | -23.7  |        | -23.6  |        | **-23.4** |        |           |
| Sand   | -16.5 | 0.3 | -16.2  | -16.8  | -15.9  | -17.1  | **-15.5** | **-17.4** | -20.4  |
| Fields | -8.8  | 0.3 | -8.5   | -9.2   | -8.2   | -9.5   | **-7.9** | **-9.8** | -12.7   |
| Trees  | -6.0  | 0.3 |        | -6.4   |        | -6.7   |        | **-7.0** | -7.4    |

**Table B4: Determination of thresholds for a sampling duration of seven months, unfiltered. Values in bold are those that have been used to calculate the threshold indicated in the last column. All values are in dB. μ – mean. σ – standard deviation.**

|        | μ     | σ   | μ+σ    | μ-σ    | μ+2*σ  | μ-2*σ  | μ+3*σ  | μ-3*σ  | Threshold |
|--------|-------|-----|--------|--------|--------|--------|--------|--------|-----------|
| Water  | -22.8 | 0.7 | -22.1  |        | -21.4  |        | **-20.7** |        |           |
| Sand   | -17.4 | 1.0 | -16.4  | -18.5  | -15.4  | -19.5  | **-14.4** | **-20.5** | -20.6  |
| Fields | -10.5 | 0.5 | **-10.0** | -11.1 | -9.5 | -11.6  | -8.9   | **-12.1** | -13.2  |
| Trees  | -6.6  | 2.0 |        | **-8.6** |        | -10.6 |        | -12.7  | -9.3    |

**Table B5: Average backscatter values from Ulaby and Dobson (1989) for C-band at VV polarization for look angles of 30° and 45°.**

|                  | 30°   | 45°   |
|------------------|-------|-------|
| Soil and rock    | -10.3 | -13.3 |
| Grasses          | -10.7 | -14.5 |
| Shrubs           | -9.7  | -11.0 |
| Short vegetation | -10.0 | -13.2 |



| | Trees | -10.8 (for 20°) | -2.3 | |
|---|---|---|---|---|

**Table B6: Confusion matrix of resultant land cover classification as obtained from SAR versus Sentinel-2 (S2) for six different months.**

| Month | Class | Sand/water (SAR) [km²] | Vegetation (SAR) [km²] | User's accuracy [%] | Producer's accuracy [%] | Overall accuracy [%] |
|---|---|---|---|---|---|---|
| Nov 18 | Sand/water (S2) | 10.79 | 1.39 | 93.8 | 88.6 | 91.3 |
| | Vegetation (S2) | 0.71 | 11.13 | 88.9 | 94.0 | |
| Nov 19 | Sand/water (S2) | 11.78 | 2.06 | 96.4 | 85.1 | 89.6 |
| | Vegetation (S2) | 0.44 | 9.78 | 82.6 | 95.7 | |
| Nov 20 | Sand/water (S2) | 11.49 | 3.69 | 93.9 | 75.7 | 81.6 |
| | Vegetation (S2) | 0.74 | 8.11 | 68.7 | 91.6 | |
| Mar 19 | Sand/water (S2) | 9.97 | 0.48 | 85.8 | 95.4 | 91.1 |
| | Vegetation (S2) | 1.65 | 11.94 | 96.1 | 87.9 | |
| Mar 20 | Sand/water (S2) | 11.02 | 0.82 | 87.7 | 93.1 | 90.2 |
| | Vegetation (S2) | 1.54 | 10.67 | 92.9 | 87.4 | |
| Mar 21 | Sand/water (S2) | 10.92 | 0.92 | 89.1 | 92.2 | 90.6 |
| | Vegetation (S2) | 1.34 | 10.85 | 92.2 | 89.0 | |


**Code availability**

The GEE code underlying the analyses of this paper is publicly available:

https://code.earthengine.google.com/eb09ee5f4635c72daca0a93d085cdbbe.

The interactive online tool implementing the findings of this paper can be accessed here:

https://code.earthengine.google.com/3ea8f1fd5d771accc621550d744a914e?hideCode=true.

**Author contribution**

JF and OF jointly conceptualized the overarching research objective and developed the methodology. JF implemented the code and prepared the manuscript with contributions from OF.


**Competing interests**

The authors declare that they have no conflict of interest.

**Acknowledgments**

We would like to thank Thomas Bernauer and Vally Koubi for helpful feedback on earlier drafts of this manuscript. Further, we would like to express our gratitude to Sudipta Hore for providing insightful background information into the erosion prediction prepared by CEGIS. We thank ESA for providing and processing the Copernicus Sentinel data [2014-2021].



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
