# Peer review of "Assessing riverbank erosion in Bangladesh using time series of Sentinel-1 radar imagery in the Google Earth Engine"

_EGUsphere, 2022_

## Referee Comment (RC1)

**Assessing riverbank erosion in Bangladesh using time series of Sentinel-1 radar imagery in the Google Earth Engine Recommendation: Minor revision**

**Recommendation:** Minor revision

**General Comments:**
This paper presents a well-designed, effectively executed, and clearly communicated project that assessed the Jamuna riverbank erosion in Bangladesh using the time series of Sentinel-1. The paper tried to contribute in the methodological aspect. The methodology is clear with the supporting documents. However, the background information in the introduction is not sufficient. So, need to address why considered the Jamuna river? (e.g., the status/dynamics and impact of Jamuna river bank erosion). Some links to the relevant study area are given below:

-https://doi.org/10.1080/15715124.2022.2068561
-https://doi.org/10.4236/gep.2017.59006
-Assessing the long-term planform dynamics of Ganges–Jamuna confluence with the aid of remote sensing and GIS (10.1007/s11069-022-05416-6)
-Prediction of fluvial erosion rate in Jamuna River, Bangladesh (10.1080/15715124.2022.2068561)
-Jamuna River Erosional Hazards, Accretion & Annual Water Discharge – A Remote Sensing & GIS Approach (doi:10.5194/isprsarchives-XL-7-W3-831-2015)

--How are the present study/methodology/findings superior (e.g., degree of accuracy, etc.) compared to the results of other studies already completed on the Jamuna river? Because, recently, there are several studies already have done on the Jamuna river! The discussion part should enlarge……..

--A number of the concluding statements included the abstract, introduction, discussion, and conclusions concerning the applicability of the work. These repetitions should be either removed or modified significantly.

--The appendices might be fix in the main body of the paper (optional)

These and other specific points to be addressed are listed in the table below.

| Actual Page number in document (Line number) | Comment |
|---|---|
| 1 (22-23) | Need references for the following statements:

 - In Bangladesh, located in one of the largest river deltas of the world (….) |
| 1 (28-30) | -Jamuna River is one of the largest braided river systems in the world, forming various channels at a total width of around 12 km (….)

 -Since the 1970s, its bank line has shifted by around 20 km (…..) |

---

## Author Comment (AC1)

**Assessing riverbank erosion in Bangladesh using time series of Sentinel-1 radar imagery in the Google Earth Engine**

Authors' response to Referee 1

| Referee comment | Authors' response |
|---|---|
| The background information in the introduction is not sufficient. So, need to address why you considered the Jamuna river? (e.g., the status/dynamics and impact of Jamuna river bank erosion). Some links to the relevant study area are given below. | Thank you for this remark. We will extend our justification of the case study selection in the introduction accordingly and add further references where needed. |
| How are the present study/methodology/findings superior (e.g., degree of accuracy, etc.) compared to the results of other studies already completed on the Jamuna river? Because, recently, there are several studies already have done on the Jamuna river! The discussion part should enlarge. | We will extend the critical reflection of our contribution in the discussion section with a focus on the added value of our radar-remote-sensing-based approach. |
| A number of the concluding statements included the abstract, introduction, discussion, and conclusions concerning the applicability of the work. These repetitions should be either removed or modified significantly. | Indeed, some sections are redundant, and we will either remove or adapt those to reduce redundancies. |
| The appendices might be fixed in the main body of the paper (optional). | In order to keep the main part of the paper compact, we chose to shift non-essential additional material to the appendices. |
| *Need references for the following statements:*

- In Bangladesh, located in one of the world's largest river deltas (….)

-Jamuna River is one of the largest braided river systems in the world, forming various channels at a total width of around 12 km (….)

-Since the 1970s, its bank line has shifted by around 20 km (…..) | Thank you for pointing this out. We will add references for the highlighted statements. |

---

## Author Response (AR1)

**Assessing riverbank erosion in Bangladesh using time series of Sentinel-1 radar imagery in the Google Earth Engine**

Authors' response to Referee 1 – Changes implemented in the manuscript

| Referee comment | Authors' response |
|---|---|
| The background information in the introduction is not sufficient. So, need to address why you considered the Jamuna river? (e.g., the status/dynamics and impact of Jamuna river bank erosion). Some links to the relevant study area are given below. | We have extended our justification of the case study selection in the introduction accordingly and added the references which you have kindly provided. |
| How are the present study/methodology/findings superior (e.g., degree of accuracy, etc.) compared to the results of other studies already completed on the Jamuna river? Because, recently, there are several studies already have done on the Jamuna river! The discussion part should enlarge. | We have extended the critical reflection of our contribution in the discussion section with a focus on the added value of our radar-remote-sensing-based approach. |
| A number of the concluding statements included the abstract, introduction, discussion, and conclusions concerning the applicability of the work. These repetitions should be either removed or modified significantly. | We either removed or adapted the mentioned sections to reduce redundancies. |
| The appendices might be fixed in the main body of the paper (optional). | In order to keep the main part of the paper compact, we chose to shift non-essential additional material to the appendices. |
| *Need references for the following statements:*

- In Bangladesh, located in one of the world's largest river deltas (….)

-Jamuna River is one of the largest braided river systems in the world, forming various channels at a total width of around 12 km (….)

-Since the 1970s, its bank line has shifted by around 20 km (…..) | We have added references for the highlighted statements. |

Authors' response to Referee 2 – Changes implemented in the manuscript

| Referee comment | Authors' response |
|---|---|
| Title: an observation– highlighting "google earth engine" in the title is not necessary; but the author can think to highlight their important study finding …. "We found that with Sentinel-1 data, erosion locations can be determined already one month after the end of the monsoon, and hence potentially earlier than using optical satellite images" for example. | Thank you for this suggestion. We believe that the main contribution of our paper is the development, implementation, and verification of a method to assess riverbank erosion along Jamuna River using satellite radar imagery. Since the Google Earth Engine is a crucial element – not so much regarding the methodology itself, but surely regarding an open and straight-forward accessibility of (1) the implementation (incl. source code) of the method, (2) handling of the large amount of Sentinel-1 SAR data (which is typically hard to handle for non-expert users), and (3) regarding reproducibility/ transferability of the approach, all readily available within the Google Earth Engine – we prefer to keep it in the title. |
| Introduction: The orientation of the Case study Bangladesh can come a bit later, so far this paper does not have any dedicated section for theoretical discussion or literature review, it this is thinkable to address here. It will help to understand the state of the art and detect the research gap and that even can relate to the motivation of the case study section and the formulation of the objective. | Our paper assesses riverbank erosion for a particular case study. As such, we believe it is appropriate to introduce the case study at the beginning of the introduction.
Concerning theory and literature review: We agree that we have not included a dedicated, in-depth theoretical section on radar scattering. To our understanding, the journal NHESS has an interdisciplinary readership for which we chose to focus on the most important aspects necessary to understand the paper (e.g. that a key advantage of using radar imagery for this particular application is its independence of daylight and weather conditions). Nevertheless, the introduction section provides an overview of the state of the art that is relevant for our case study.
Having said that, we have reworked the introduction to even better highlight the research gap with respect to the state of the art. |
| Section 2 "Method and data": some of the sub-section headings are identical to section 3 "Results" e.g. 2.3 same as 3.1, 2.4 same as 3.2. This is completely confusing and contributes to poor readability. It is recommended to revise the sub-section headings in the "Results" section. They should be more declarative rather than general about the particular highlights or findings of the section. | We have adapted the headings to distinguish them from those in the methods section. |
| Some terminology can be more clarified – e.g. "land" - does it refer to open space / agricultural land/ forest? | "Land" refers to agricultural land. We have clarified this in the paper. |

| | |
|---|---|
| The URL link for code and tool needs to be presented in a standard format and with Meta-description (repository like platform Zenodo with DOI may be an option) and push the access link URL, DOI in them in the annexe of the paper only refer them in the original text. | We have uploaded the code and video to Zenodo and added the DOI to the section "Code availability". |
| After opening the given link - in the current state – it makes to rethink the author's statement "the code and tool developed in this study might be of interest to both policymakers and practitioners working in the fields of disaster risk management and communication" (SECTION 430) | Thank you for this feedback. We agree that the source code of the tool might not be of interest to policymakers and practitioners. We have removed "the code" from the statement you highlighted.

Concerning the usability of the tool, we believe that the tutorial which we have recorded and published on YouTube will allow policy-makers and practitioners to use the tool and to replicate the results of our analysis at low effort. |
| The paper focused only on the physical aspect by quantification changes and intensity of erosion; however, it will be necessary to shed light on some discussion and policy implications – how these results can be fed to the other research direction for example socio-economic dimensions. | Thank you. Indeed, the results of the study are directly used in a larger social science research project studying the adaptation of the population living along Jamuna River to the riverbank erosion: https://p3.snf.ch/Project-185210. We have extended the conclusions section to clarify the potential use of our results for social scientific research. |
| So far, the data processing task allows for to production of indicators in time series and spatial scale; the reader may also expect - what is the scope to do some predictive analytics in the future research scope. | We agree that prediction of erosion would obviously be very useful. However, we think that a meaningful erosion prediction involves a whole set of additional (previously unknown) parameters (prediction of precipitation, river discharge, numerical modelling of fluvial processes etc) and predictive methods, which are simply out of the scope of the research presented in our manuscript. We rather see the erosion classification obtained in our work as a useful and readily available INPUT to predictive approaches.

Examples of such approaches include a probabilistic prediction of riverbank erosion based on past events. Please note that e.g. Dhaka-based CEGIS provides a probabilistic prediction of riverbank erosion for the Jamuna River each year.

Then, if long enough time series are available, machine learning might help to predict erosion (but also this approach would be merely classification-data-driven and would not include the strongly variable unknown future weather / discharge etc parameters). |

---

## Author Response (AR2)

**Assessing riverbank erosion in Bangladesh using time series of Sentinel-1 radar imagery in the Google Earth Engine**

Authors' response to Referee 1

| Referee comment | Authors' response |
|---|---|
| Discussion........
In terms of accuracy, our algorithm performed satisfactorily when compared to an approach based on optical images.

Please provide an example for the above-mentioned statement (compared to other studies of the Jamuna River). Otherwise, you can't say!

So, you should either include information or rewrite it. | Thank you for this remark. The statement which you mention does not refer to other studies which have been performed on the Jamuna River using optical data, but rather to section 3.4 of our own manuscript, where we compare our radar-based approach to a classification resulting from using optical images. We have added a reference to section 3.4 in the discussion section, to clarify what we refer to. |